# Dimensionality Matters: Exploiting UV-Photopatterned 2D and Two-Photon-Printed 2.5D Contact Guidance Cues to Control Corneal Fibroblast Behavior and Collagen Deposition

**DOI:** 10.3390/bioengineering11040402

**Published:** 2024-04-19

**Authors:** Cas van der Putten, Gozde Sahin, Rhiannon Grant, Mirko D’Urso, Stefan Giselbrecht, Carlijn V. C. Bouten, Nicholas A. Kurniawan

**Affiliations:** 1Department of Biomedical Engineering, Eindhoven University of Technology, P.O. Box 513, 5600 MB Eindhoven, The Netherlands; 2Institute for Complex Molecular Systems, Eindhoven University of Technology, P.O. Box 513, 5600 MB Eindhoven, The Netherlands; 3MERLN Institute for Technology-Inspired Regenerative Medicine, Department of Cell Biology-Inspired Tissue Engineering cBITE, 6229 ER Maastricht, The Netherlands

**Keywords:** contact guidance response, collagen alignment, UV-based protein patterning, two-photon polymerization of topographies, cornea, corneal fibroblasts

## Abstract

In the event of disease or injury, restoration of the native organization of cells and extracellular matrix is crucial for regaining tissue functionality. In the cornea, a highly organized collagenous tissue, keratocytes can align along the anisotropy of the physical microenvironment, providing a blueprint for guiding the organization of the collagenous matrix. Inspired by this physiological process, anisotropic contact guidance cues have been employed to steer the alignment of keratocytes as a first step to engineer in vitro cornea-like tissues. Despite promising results, two major hurdles must still be overcome to advance the field. First, there is an enormous design space to be explored in optimizing cellular contact guidance in three dimensions. Second, the role of contact guidance cues in directing the long-term deposition and organization of extracellular matrix proteins remains unknown. To address these challenges, here we combined two microengineering strategies—UV-based protein patterning (2D) and two-photon polymerization of topographies (2.5D)—to create a library of anisotropic contact guidance cues with systematically varying height (H, 0 µm ≤ H ≤ 20 µm) and width (W, 5 µm ≤ W ≤ 100 µm). With this unique approach, we found that, in the short term (24 h), the orientation and morphology of primary human fibroblastic keratocytes were critically determined not only by the pattern width, but also by the height of the contact guidance cues. Upon extended 7-day cultures, keratocytes were shown to produce a dense, fibrous collagen network along the direction of the contact guidance cues. Moreover, increasing the heights also increased the aligned fraction of deposited collagen and the contact guidance response of cells, all whilst the cells maintained the fibroblastic keratocyte phenotype. Our study thus reveals the importance of dimensionality of the physical microenvironment in steering both cellular organization and the formation of aligned, collagenous tissues.

## 1. Introduction

Corneal blindness is the most common visual impairment worldwide, and is mostly treated via the transplantation of corneal donor tissue. With only one cornea available for every seventy needed, there is a severe global shortage of donor tissue. As a consequence, more than 12 million patients remain untreated, indicating the urgent need for therapeutic alternatives for corneal regeneration [1]. As the outermost part of the eye, the cornea is responsible for the focusing of all incoming light, and is thus a crucial tissue that enables proper vision. This function is enabled by the precise organization of cornea-specific cells and extracellular matrix (ECM). This is especially true for the corneal stroma, the thickest and central layer of the cornea, which consists of densely packed, anisotropic collagen bundles arranged in orthogonally oriented lamellae with corneal keratocytes dispersed throughout. This orthogonal organization not only allows light to properly pass through the cornea, but also provides a specific biomechanical and topographical environment to the resident cells, corneal keratocytes [2]. The corneal ECM predominantly consists of fibrous collagens type I and V, which are tightly packed with well-defined and highly conserved inter-fibril spacing. Corneal keratocytes are sparsely populated throughout the collagen lamellae, and are responsible for the production and maintenance of this highly organized stromal tissue [3]. In the healthy adult cornea, keratocytes remain quiescent and typically adopt a dendritic shape [4]. However, upon disease or injury, keratocytes can quickly lose their quiescent phenotype and dendritic shape to become activated (myo)fibroblast-like cells that no longer maintain the precise organization of the tissue. Instead, these cells produce aberrant ECM components, leading to a disorganized, opaque ECM that eventually impairs vision [5]. In order to repair the dysfunctional stromal ECM and restore vision, controlled regeneration by the precisely organized keratocytes is necessary [5,6].

Multiple bioengineering approaches have been used to mimic the organization of cells and ECM in the stromal tissue, and thereby restore the function of the stromal ECM [7]. A widely used approach is to harness the ability of adherent cells to align along their direct environment, a so-called “contact guidance” response. To this end, contact guidance has been extensively recapitulated and studied using a variety of laboratory setups, for example, using highly defined in vitro substrates presenting grooves (nanoscale [8,9,10,11,12] and microscale [10,11,13,14,15,16,17,18,19,20,21,22]), lattices [13], micropits [13,22], anisotropic liquid crystalline networks [23], and electrospun fibers [24,25,26,27,28,29]. Depending on the precise dimensions of the (anisotropic) nano- and micro-scale cues, activated keratocytes can spread out and align along the direction of the environmental cue, after which the newly formed collagen is deposited along the same direction [24,30].

Despite the number of studies demonstrating the power of contact guidance cues for steering cell behavior, researchers often focus on a single or limited number of pattern dimension(s) that to some extent resembles the organization of the native environments. Furthermore, most studies mainly focus on the width of contact guidance cues. Excitingly, in recent years, more attention has been paid to the role of topography height, which is increasingly appreciated as an important parameter responsible for distinct cell behaviors [31,32,33]. Indeed, from previous in vitro research, it is known that cue dimensionality (e.g., topography width and height) can have a major influence on cell behavior [34,35]. Since most existing in vitro contact guidance cues are oversimplified compared to the native cellular environment, it is therefore crucial to investigate the relevance of cue dimensionality on keratocyte alignment and collagen deposition. To our knowledge, there have not been systematic investigations into the impact of the dimensionality of contact guidance cues on the morphological organization of corneal keratocytes and the production of aligned ECM components at the mesoscale, which is a vital step for corneal tissue engineering. Here, we undertook an in-depth characterization of the effects of the width and height of contact guidance cues on fibroblastic primary corneal keratocyte behavior, also referred to as activated keratocytes, and collagen alignment. Guided by the organization of decellularized human corneal stroma, we generated a library of surface patterns. We systematically simplified the native 3D physical cues in a stepwise approach to 2.5D protein-coated substrate topographies [36,37] and 2D protein patterns [38] to elucidate the role of cue dimensionality on cell behavior [39] and collagen alignment. We examined the cellular contact guidance response and morphology after 1 day, and the orientation of both cells and collagens after 14 days. This systematic approach to understand the impact of contact guidance cues allows us to pinpoint the dimensional needs of activated, collagen-producing keratocytes, and aids tissue engineers in the development of more representative microenvironments for the in vitro culture of primary (fibroblastic) keratocytes.

## 2. Materials and Methods

### 2.1. Scanning Electron Microscopy Imaging of Corneal Stroma

All human corneal tissues used in this study were obtained from leftover human corneoscleral transplant material from Descemet Membrane Endothelial Keratoplasty surgery. Tissues were obtained from the Cornea Department of the ETB-BISLIFE Multi-Tissue Center (Beverwijk, The Netherlands). Consent from the next-of-kin of all deceased donors was obtained, and the research was performed in compliance with the tenets of the Declaration of Helsinki. The cross-section of the tissue was prepared as described previously [40]. Briefly, the cornea was decellularized in 10% sodium hydroxide solution (Merck, Darmstadt, Germany) at room temperature for 6 days followed by dehydration in a graded series of ethanol (70–90–100%, Merck, Darmstadt, Germany). After fixation with 1% osmium tetroxide (Electron Microscopy Sciences, Hatfield, PA, USA), the specimen was cut on the Y-axis with a single-edge razor blade (Tmall). After coating with 10 nm gold in a sputter coater (108auto, Cressington, Watford, United Kingdom), the corneal Y-plane was imaged using Scios DualBeam scanning electron microscopy (SEM, 5 kV, 1000× magnification, Thermo Fisher Scientific, Waltham, MA, USA).

### 2.2. Design of Contact Guidance Patterns

To allow us to mimic the dimensionalities of native corneal stroma, the architecture of human cornea was examined. The dimensions of the voids between orthogonally oriented lamellae (Appendix A) were measured on the SEM images using Fiji (Version 1.52p, NIH) [41]. Subsequently, anisotropic contact guidance patterns were designed with gradually increasing widths (W) and accompanying spacing of 5, 10, 20, 50, and 100 µm (Figure 1A). The height (H) of the 2.5D cues varied from 20 μm, to 10, 5, and 2.5 μm. The 2D contact guidance cues were designed similarly, with similar cue widths and no height. The nomenclature used for all contact guidance cues is described in Table 1.

### 2.3. 2D Contact Guidance Cues

Two-dimensional contact guidance cues were realized as adhesive protein line patterns by means of a UV-photopatterning approach. Patterning of glass substrates was then performed using the light-induced molecular adsorption of proteins (LIMAP) method using PRIMO equipped with Leonardo software (version 4.13, Alvéole, Paris, France) [38]. First, glass slides (Menzel-Gläser, 13 and 32 mm Ø, #1) were treated with O_2_-plasma (30 s at 20 W, K1050X, Quorum, Laughton, UK) to activate the substrate. Immediately after plasma treatment, poly-L-lysine (PLL, 0.01%, P4707, Sigma-Aldrich, Burlington, MA, USA) was incubated for 30 min at room temperature, followed by 3× washing with N-2-hydroxyethylpiperazine-N-2-ethane sulfonic acid (0.1 M, HEPES, 8.0 < pH < 8.5, 15630080, Gibco, Waltham, MA, USA) and incubation with mPEG-succinimidyl valerate (mPEG-SVA, 50 µg/mL in 0.1 M HEPES (8.0 < pH < 8.5), MW 5000 Da, Laysan Bio, Inc., Arab, AL, USA) for 60 min at room temperature. After incubation with mPEG-SVA, samples were washed 3× using PBS (P4417, Sigma) and transferred to the PRIMO setup for UV-photopatterning. Here, photoinitiator (4-benzoylbenzyl-trimethylammonium chloride (PLPP), Alvéole, Paris, France) was pipetted on the passivated glass slide, after which the digital pattern was projected on top of the glass slide using UV light with a dose of 1000 mJ/mm^2^. After patterning, substrates were transported to sterile culture cabinets and washed with ethanol (70% *v*/*v*, 0005250210BS, Biosolve, Dieuze, France) and 5× with sterile PBS before being incubated with fluorescently labeled gelatin (fluorescein-gelatin, 0.01%, G13187, Invitrogen, Waltham, MA, USA) for 15 min at 37 °C. Finally, the substrates were washed extensively with sterile PBS to remove excess fluorescein-gelatin.

### 2.4. 2.5D Contact Guidance Cues

The 2.5D contact guidance cues were produced on indium tin oxide (ITO)-coated soda lime glass slides (25 mm × 25 mm × 0.7 mm, Nanoscribe GmbH & Co. KG, Eggenstein-Leopoldshafen, Germany). Glass slides were first rinsed with acetone, distilled water, and isopropanol (VWR, Netherlands), after which all substrates were blow-dried with nitrogen and activated using oxygen plasma (2 min at 70 W, oxygen flow at 2 sccm, Diener electronic Femto plasma etcher). Next, the substrates were placed in a glass petri dish with a 150 μm droplet of 3-(trimethoxysilyl)propyl methacrylate (98%, Sigma-Aldrich, Burlington, MA, USA) and put in a vacuum oven overnight (180 mbar, 110 °C, Vacutherm, Thermo Fisher Scientific, Waltham, MA, USA). Functionalized ITO-coated glass slides were stored at 4 °C until further use. Line and space patterns to fabricate 2.5D contact guidance cues were designed using computer-aided design (CAD) software (SolidWorks 2015), and exported to an STL format, which was further processed by Nanoscribe DeScribe software to optimize the printing process for two-photon polymerization (2PP) lithography. The contact guidance cues were printed in low-fluorescence IP-Visio resin (Nanoscribe GmbH & Co. KG, Eggenstein-Leopoldshafen, Germany) using a 2PP lithography setup (Photonic Professional GT, Nanoscribe GmbH & Co. KG, Eggenstein-Leopoldshafen, Germany) on the functionalized ITO-coated glass slides. Once the printing process was finished, the slides were developed in Mr-Dev 600 (Micro Resist Technology GmbH, Berlin, Germany) for 20 min, followed by 30 s in methoxy-nonafluorobutane (Novec 7100, 3M) and gentle blow-drying. Before cell culture, substrates were washed 3× with ethanol (70%, VWR) and 3× with PBS, and incubated with 200 μL fluorescein-gelatin (0.01%, G13187, Invitrogen) for 15 min at 37 °C. After incubation, excess gelatin was aspirated and substrates were extensively washed with PBS.

### 2.5. Characterization of 2D and 2.5D Contact Guidance Substrates

Height maps for all substrates were obtained using confocal laser scanning profilometry (VK-X200, Keyence, Osaka, Japan). The profilometry data were processed using the Keyence Multifile Analyzer software and plotted using MATLAB (version 2020a, The Mathworks, Natick, MA, USA). The 2D contact guidance cues were characterized using confocal microscopy (TCS SP8X confocal microscope, 10× 0.4 NA objective, Leica, Wetzlar, Germany). The 2.5D contact guidance cues were inspected using scanning electron microscopy (SEM) (JSM-IT200 InTouchScope, Jeol, Tokyo, Japan) to qualitatively evaluate the fabrication process. For SEM imaging, samples were sputter-coated with a 10 nm layer of gold (108auto, Cressington, Watford, UK) and affixed to SEM chucks using double-sided carbon tape.

### 2.6. Cell Isolation and Culture on Contact Guidance Substrates

Primary human keratocytes were isolated from donated human cornea (Euro Tissue Bank, ETB-BISLIFE Multi-Tissue Center, Beverwijk, The Netherlands). Endothelial and epithelial cells were removed from the tissue by means of scraping with a flat razor blade. Any limbal and scleral tissues were removed by using an 8 mm biopsy punch in the center of the cornea. Using a sterile blade, the remaining tissue was minced into pieces of approximately 1 mm × 1 mm. The pieces were placed in a 15 cm petri dish and incubated in a cell culture incubator (37 °C, 5% CO_2_) for 6–8 h in 7 mL of digestion medium (Dulbecco’s Modified Eagle Medium/Nutrient Mixture F-12 Glutamax (DMEM F12, Gibco, Waltham, MA, USA) supplemented with 5% fetal bovine serum (FBS, Sigma–Aldrich, Burlington, MA, USA), 1 mM ascorbic acid (ASAP, Sigma–Aldrich, Burlington, MA, USA), 10 IU/mL penicillin (Gibco, Waltham, MA, USA), 0.1 mg/mL streptomycin (Gibco, Waltham, MA, USA), 1 mg/mL collagenase I (Thermo Fisher Scientific), and 0.1% bovine serum albumin (BSA, Sigma–Aldrich, Burlington, MA, USA)), until 90% of the tissue was digested. Digestion was carried out up to 12 h, since digestion of more than 16 h is detrimental for cells [42]. When the tissue was easily malleable with pipette tips and minimal force, digestion was considered to be 90%. Here, cells were present in the media, whereas ECM particulates remained present. After incubation, large pieces of undissociated material were removed by first using a 100 µm cell strainer, followed by a 40 µm cell strainer. The filtrate was diluted 3× with PBS and centrifuged (350 g, 7 min) to separate all cells. The supernatant was removed, and the pellet was washed in 3 mL PBS, followed by an additional centrifugation step (350 g, 7 min). Collected cells were counted and seeded (10,000 cells/cm^2^) in expansion medium (Dulbecco’s Modified Eagle Medium/Nutrient Mixture F-12 Glutamax (DMEM F12, Gibco, Waltham, MA, USA)) supplemented with 5% fetal bovine serum (FBS, Sigma Aldrich), 1 mM ascorbic acid (ASAP, Sigma–Aldrich, Burlington, MA, USA), 10 IU/mL penicillin (Gibco, Waltham, MA, USA), and 0.1 mg/mL streptomycin (Gibco, Waltham, MA, USA)). Upon seeding on substrates, 250 µL of cell suspension (40,000 cells/mL) was incubated on control (fluorescein-gelatin-coated glass, 0.01%, G13187, Invitrogen, Waltham, MA, USA), 2.5D, and 2D substrates for 1 h at 37 °C and 5% CO_2_. After 1 h, all samples were gently rinsed with medium to remove any unattached cells, after which an excess of medium was added. Since the same medium composition (including 5% FBS) was used throughout the entire study, keratocytes were assumed to be ECM-producing, fibroblast-like cells. Samples were cultured for 1 day and 7 days with medium changes every 3 days. Keratocytes from multiple donors were used up to passage 4. Live-cell imaging was performed using a CytoSMART Lux3 BR (Axion BioSystems B.V., Eindhoven, The Netherlands).

### 2.7. Staining and Imaging of Keratocytes and Extracellular Matrix Proteins

All fibroblastic keratocytes on the cell culture substrates were fixed for 15 min at room temperature using 3.7% paraformaldehyde (formalin 37%, 104033.1000, Merck, Darmstadt, Germany). After washing with PBS, samples from day 1 were stained for nuclei using 4′,6-diamidino-2-phenylindole dihydrochloride (DAPI, D9542, Sigma-Aldrich, Burlington, MA, USA), F-actin using Phalloidin-Atto647 (65906, Sigma-Aldrich, Burlington, MA, USA), and vinculin (1st: 1:600 mouse anti-vinculin IgG1 antibody, Sigma, V9131, 2nd: 1:300 goat anti-mouse IgG1-Alexa555, Molecular Probes, A21127). Seven-day cultures were stained for nuclei (DAPI), F-actin (Phalloidin-Atto647), and collagen (CNA35-mCherry [43], manufactured in-house). A TCS SP8X confocal microscope with 10× 0.4 NA and 40× 0.95 NA objectives (Leica, Wetzlar, Germany), and a Dmi8 epi-fluorescent microscope with 10× 0.4 NA objective (Leica, Wetzlar, Germany) were used to obtain Z-stacks (5 µm Z-spacing) of all samples. 

### 2.8. Quantification of Cell Morphological Parameters

Initial analysis of microscopy data was performed using Fiji to create maximum-intensity projections of acquired Z-stacks. Subsequently, CellProfiler (4.2.1, Broad Institute, Inc., Cambridge, MA, USA) [44] was used to analyze morphological parameters of primary keratocytes on day 1 samples. To this end, the projected cell and nuclear areas were determined by the number of pixels, from each object on maximum-intensity projections, and were converted into an area (µm^2^). The minor and major axis length was determined as the minor and major axis of an ellipse that contains the object. Eccentricity as a measure for the roundness of objects was calculated as the ratio between the major and minor axis of that same ellipse, with 0 representing a perfect circle and 1 a perfectly straight line. The organization of both cells and collagen on 7-day cultures was analyzed using an in-house-developed MATLAB fiber analysis tool, FOAtool, to determine F-actin and collagen fiber directionality by means of Frangi vesselness [45]. Herein, F-actin (phalloidin) signals were analyzed as a representative measure for cell orientation. The orientation of cells and collagen was quantified with respect to the direction of the patterns, with 0° representing perfect alignment. Cell morphology parameters were not characterized at later time points, as the cells formed a dense layer where individual cells could no longer be identified.

### 2.9. Verification of Keratocyte Phenotype Using Real-Time Polymerase Chain Reaction

Two-step reverse transcription–quantitative polymerase chain reactions (RT-qPCR) were used to assay for markers of interest expressed by keratocytes cultured on selected substrates (H0W10 and H20W10) for 7 days. Q-PCR results were analyzed and standard deviation was propagated using the 2^−ΔΔCt^ method [46]. Cells and scaffolds were homogenized using Trizol (Life Technologies, Waltham, MA, USA). RNA was isolated following chloroform-mediated phase separation and purified using a RNeasy kit (Qiagen, Hilden, Germany) as per manufacturer’s instructions. cDNA was synthesized using an iScript™ cDNA Synthesis Kit (Biorad, Hercules, CA, USA), according to manufacturer’s instructions. Gene expression levels were normalized using expression of the housekeeping gene Peptidylprolyl isomerase A (PPIA) and presented as a relative expression compared to keratocytes cultured on flat glass substrates. Aldehyde dehydrogenase 3 Family member A1 (ALDH3A1), Lumican, Keratocan, Alpha smooth muscle actin (α-SMA), Cluster of differentiation 90 molecule (CD90), vimentin, and Cluster of differentiation 34 molecule (CD34) were investigated. Exons spanning forward and reverse primers (Sigma) were designed using the free-to-access NCBI PrimerBLAST and are detailed in Appendix A [47].

### 2.10. Statistical Analysis

Relative frequency distributions (orientation) were calculated using Origin (version 2018, OriginLab, Northampton, MA, USA), and statistical differences between conditions were determined using the Kolmogorov–Smirnov test in MATLAB. All other morphometric data were first subjected to outlier removal (based on median, MATLAB), followed by a normality test and the Wilcoxon rank-sum test for statistical analysis in MATLAB. Data plots are visualized using Origin, and the heatmaps using Heatmapper [48]. All data were collected from a minimum of three replicates per condition. All data presented are the mean ± standard error of the mean, unless indicated otherwise. The 2^−ΔΔCt^ values from the RT-qPCR assay were subjected to a Shapiro–Wilk test for normality, followed by an ANOVA or Kruskal–Wallis test with Dunn’s multiple comparison analysis in MATLAB. *p*-values were as follows: * *p* ≤ 0.05, ** *p* < 0.01, *** *p* < 0.001, and **** *p* < 0.0001.

## 3. Results and Discussion

### 3.1. Fabrication of Contact Guidance Substrates

The extracellular environment can steer keratocytes into a behavior indicative of the normal and healthy healing response, leading to the formation of aligned collagen lamellae and eventually tissue regeneration and restoration of functionality. Within their native environment, dendritically shaped keratocytes are organized in linear arrays aligned with, and scattered between, collagen lamellae. In vivo, this complex 3D microniche both supports and is maintained by keratocytes. To elucidate the natural contact guidance cues provided by the 3D environment, we used the dimensions provided by SEM imaging of a decellularized donor cornea. To map the dimensions of anisotropic 2.5D and 2D contact guidance cues (Figure 1A), the dimensions of the voids between orthogonally oriented lamellae in the decellularized cornea were measured, where keratocytes likely reside in the healthy tissue (Appendix A), and it was found that the average lateral and vertical dimensions of the voids are 25 ± 13 μm and 9 ± 2 μm, respectively. Next, line arrays were designed in a format that simultaneously enables the assessment of various cue widths on cell behavior, combining contact guidance cues of W = 5, 10, 20, 50, and 100 µm for all individual heights (Figure 1A). Using 2PP and UV-photopatterning, highly reproducible cell culture substrates containing arrays of contact guidance cues were produced. Cue dimensionality was verified and proven accurate and consistent across all substrates (Figure 1B,C). To fairly compare any following cell culture results between the 2.5D and 2D contact guidance cues, an identical surface coating with gelatin-fluorescein was applied on all samples. Gelatin was used as a collagen-mimicking surface coating on all samples to avoid uncontrolled contact guidance responses that are normally induced by fibrillar collagens.

### 3.2. Orientation of Fibroblastic Keratocytes Is Affected by 2.5D and 2D Contact Guidance Cues

We firstly investigated how the various cue dimensions influence the orientation of activated keratocytes. Confocal microscopy and orientation analysis of single cells after 24 h of cell culture revealed that fibroblastic keratocytes exhibit a strong orientation response in the presence of 2.5D and 2D contact guidance cues (Figure 2 and Appendix A, Appendix A). The fractions of aligned cells (Figure 2A) on all contact guidance substrates were significantly higher than on flat, control substrates (aligned fraction of 0.126 ± 0.003), consistent with previous studies showing the keratocyte response on patterned substrates [10,11,13,14,15,16,17,19,20,21,22]. Keratocytes on all 2.5D cues showed a stronger alignment (Figure 2C–F) compared to on 2D (Figure 2B), providing a first indication that topography height directly influences the orientation of fibroblastic keratocytes. Moreover, cell alignment on 2D contact guidance cues of various widths followed a different trend than on 2.5D cues. While the highest alignment of activated keratocytes on 2D samples was observed on 20 μm wide patterns (Figure 2B), alignment in 2.5D environments was highest on narrow (H5, H10, H20) topographies (Figure 2C–F). Despite the minor (2.5 μm) height difference between H0W5 and H2.5W5, the orientation of cells was increased almost four-fold (Figure 2B,C). This effect was strongest on narrow cues, such as W5 and W10, compared to the wider cues, such as W50 and W100. The relevance of topography height in activated keratocytes shown here corroborates recent findings with other cell types [32].

### 3.3. Cue Dimensionality Directly Influences Morphology and Focal Adhesions of Fibroblastic Keratocytes

As shown in the previous section, in the presence of contact guidance cues, fibroblastic keratocytes notably changed their adhesion morphology. To further investigate the effect of cue dimensions on the morphological change, additional cell and nuclear morphometric properties such as projected area, eccentricity, and major and minor axis lengths were analyzed (cell bodies: Figure 3 and Appendix A; nuclei: Appendix A). Overall, heatmaps revealed that cell and nuclei morphology follow the same trend on all substrates. Although both topography height and width influenced cell morphology, the impact of topography height was more pronounced compared to the cue “width” (cell bodies: Figure 3B,E and Appendix A; nuclei: Appendix A). Another important observation was that the cell projected area for the same width but different heights showed biphasic response, i.e., an increase in area with the increase in topography height up to 5 μm followed by a decrease in area with the further increase in topography height until 20 μm (Figure 3B). These data suggest that fibroblastic keratocytes expand in the vertical dimension on topographies with heights greater than 5 µm. In contrast, cell eccentricity, which reflects the degree of cell elongation, monotonically increased with the increasing pattern height (Figure 3C). The major axis length showed a similar biphasic response on topography height as cell projected area (Figure 3D). This can also explain the increase in cell eccentricity at heights equal to and smaller than 5 µm. Interestingly, the decreasing major axis length with increasing topography height at heights greater than 5 µm is accompanied by a general decrease in minor axis length (Figure 3E), explaining the further increase in cell eccentricity at this height range. 

Further quantification of cell adhesion, shape, and morphology indicates an interesting trend that the dimensionality of the patterns invoked different adhesion strategies of the activated keratocytes (Figure 4). Cells were able to spread across multiple 2.5D and 2D cues when the height was either 0 or 2.5 µm (white arrows in Figure 4), but became increasingly confined in between the higher anisotropic cues (H5, H10, and H20, red arrows in Figure 4). On taller cues, most fibroblastic keratocytes adhered to either the top or the bottom surface. Additionally, cells were observed to adhere to the sides of the taller features (yellow arrows in Figure 4), leading to a decrease in projected area with increasing topography height. Therefore, in this case, the projected area is expected to be an underestimation of the total adhesion area of the cells. On substrates with topography heights smaller than 5 µm, fibroblastic keratocytes showed adhesion behavior previously also observed by myofibroblasts, bridging multiple cues [49]. The presence of taller contact guidance cues induced the confinement of keratocytes. On these substrates, fibroblastic keratocytes could not bridge multiple cues, resulting in adhesion in between two individual ridges. This adhesion behavior was mainly seen on the substrates with cues between 5 and 20 µm wide, in which the spacing between two subsequent cues varied between 5 and 20 μm. In the case of 50 and 100 µm wide patterns, the size of individual cells limited the ability to adhere to multiple topographic ridges at the same time. As a consequence, cells adhered to the sides of a single ridge. Overall, the distinct adhesion strategies of fibroblastic keratocytes stress the importance of both the lateral and vertical dimensions of substrates presenting contact guidance cues. 

### 3.4. Maintaining the Cellular Phenotype and Steering Collagen Formation Using Contact Guidance Cues

In addition to the cell morphology and adhesion behavior, it is known that the genetic profile of keratocytes can also be influenced by the physical environment [10,14,17,50,51,52]. Damage to the highly organized collagen fibers in the native ECM activates keratocytes in the corneal stroma and leads to unwanted fibrosis, resulting in scar tissue formation [6]. It is crucial to understand how such fibrosis can be prevented and how to culture functional keratocytes in vitro. Therefore, the potential of contact guidance cues that can prevent the formation of fibrotic cells even under the harsh isolation process and culture conditions involving soluble cues promoting fibrotic behavior, such as FBS, was investigated [53]. Given that the impact of topography height on cellular behavior was more significant than that of cue width (cell bodies: Figure 3B,E and Appendix A; nuclei: Appendix A), we fabricated two new culture substrates comprising entirely 10 μm wide cues, with heights of either 0 or 20 μm. Using these two conditions (H0W10 and H20W10), a reverse transcription RT-qPCR to assess the impact of topography height on cellular phenotype was conducted. Fibroblastic keratocytes cultured on H0W10 contact guidance cues showed no change in expression of all keratocyte markers, including ALDH3A1, a protective corneal crystalline, Keratocan, one of the major proteoglycans of the stroma, CD34, an alternate keratocyte marker downregulated in injured corneas, and Lumican, which regulates collagenous matrix assembly as a keratan sulfate proteoglycan (Figure 5A). There was a significant increase for vimentin, which is a major structural intermediate filament (type III) protein expressed in injured corneal fibroblasts compared to keratocytes cultured on flat substrates (Figure 5B). On H20W10 topographies, keratocytes showed a significantly lower expression of ALDH3A1 (Figure 5A). Since the expression of most markers remained unchanged upon culture on the contact guidance substrates, no phenotypic switch was found in these conditions.

Aligned adhesive cells secrete collagen through fibripositors, mainly located at the rear and front poles of the cells [54]. To investigate the influence of the defined surface patterns on collagen fiber alignment, the activated keratocytes were cultured for 7 days on contact guidance cue presenting substrates. On all substrates, fibroblastic keratocytes were able to produce dense fibrous networks of collagen (Figure 6). After 7 days of culture, in the absence of any contact guidance cues, no global organization of cells was present, and the deposited collagen network was observed (Figure 6A). The 2D contact guidance cueswith widths equal to and smaller than ~50 μm were completely covered with fibroblastic keratocytes, including the non-adhesive space in between lines (Figure 6A, Appendix A). However, on substrates with lines wider than 50 μm, cells only adhered on top of the protein pattern, thereby avoiding adhesion on any non-adhesive area. Consequently, the deposited collagen network was also only present on top of the adhesive, protein-patterned lines (Figure 6A). The 2.5D contact guidance cues presenting substrates with heights of 2.5 μm and 5 μm induced the fibroblastic keratocytes to proliferate in between and on top of the cues (Figure 6B), where cells mostly stayed in between the cues on substrates with cues 10 μm and 20 μm tall. Therefore, the deposited collagen was mainly located in between the cues 10 μm and 20 μm tall. Moreover, the result showed that the collagen fibers deposited by the cells were aligned parallel to the cells and oriented in the direction of contact guidance cues (Figure 6C,D, Figure 7, and Appendix A). This orientation was uniform as topography height increased (Figure 7B–F). For both cells (Appendix A) and the newly deposited collagen (Figure 7E,F), the highest degree of alignment was found on the taller cues 10 µm wide due to the confinement of cells between cues. 

As can be seen in the orientation distributions of collagen on 2D substrates (Figure 7B and Appendix A), activated keratocytes collectively oriented in one direction, albeit with a clockwise rotation. This behavior was not observed on the substrates with topographical contact guidance cues, as the physical 2.5D environment most probably limited the collective rotation of cells and collagen during culture. The clockwise rotation of both activated keratocytes and the deposited collagen is consistent with in vivo observations during stromal development in chickens [55].

## 4. Conclusions

For many patients, loss of vision is a direct consequence of a disorganized corneal stroma. In healthy tissue, collagen fibers are organized in tight, orthogonal bundles called lamellae. Keratocytes are responsible for the production and maintenance of the collagenous ECM, and are influenced by the anisotropic collagen bundles in the cellular environment. Taking inspiration from the architecture of the human corneal stroma, we have distilled complex physical cues into simplified anisotropic 2.5D and 2D contact guidance cues in order to elucidate the impact of cue guidance on (activated) corneal keratocyte behavior. These cues were precisely engineered with diverse widths and heights, recapitulating key morphological features of the inherent architecture of the stroma. Our findings demonstrate that the dimensional features of these contact guidance cues profoundly steer the orientation of fibroblastic keratocytes. For instance, a 2.5 μm height difference between 2D and 2.5D cues led to a remarkable increase in cell orientation, up to four-fold. Furthermore, in the presence of topographical variation in height, the alignment effect was particularly pronounced on narrower cues, without inducing any phenotypic switching. Notably, the orientation of collagen deposition after 7 days of culture was also profoundly influenced by the dimensions of the contact guidance cues. This effect was distinct, with orientation increasing up to three-fold in response to elevated topographical heights. Collectively, our findings showed the potential of employing simplified 2.5D and 2D contact guidance cues to effectively modulate cell behavior, subsequently influencing collagen deposition. Overall, the height of the surface topographies emerges as a pivotal factor. Reducing the contact guidance cues to 2D and eliminating the height component could potentially result in the absence of specific instructive cues crucial for activated keratocyte behavior. While 2D cues alone exhibit competence in cell guidance, it becomes evident that the synergistic effect of surface topography with its associated height holds significant importance. This systematic approach, aimed at comprehending the intricate influence of contact guidance cues, enables us to precisely identify the dimensional prerequisites governing the behavior of fibroblastic keratocytes. This, in turn, assists tissue engineers in designing more faithful microenvironments for the in vitro culture of (activated) keratocytes. Future research should be directed to exploiting these insights to develop three-dimensional engineered corneal tissues and examining how their functions are influenced by the cell and tissue architecture.

## Figures and Tables

**Figure 1 bioengineering-11-00402-f001:**
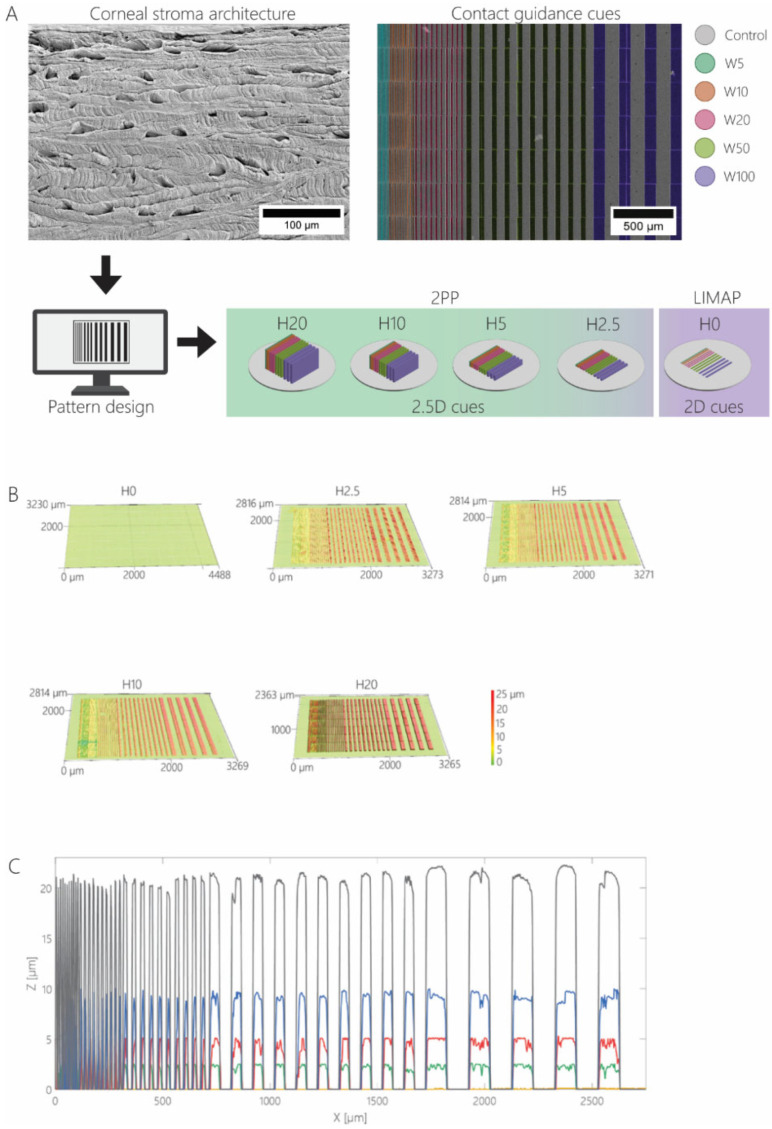
(**A**) The simplification of 3D corneal stromal architecture into line patterns, followed by the fabrication of cell culture substrates with 2.5D and 2D contact guidance cues using 2PP and LIMAP, respectively. All 2.5D cues contained pattern widths of 5, 10, 20, 50, and 100 μm combined on a single substrate. An example of a false colored scanning electron micrograph shows the top-view of a substrate with 20 µm tall ridges and a width of 5 μm (blue), 10 μm (orange), 20 μm (pink), 50 μm (green), and 100 μm (purple). The height of the topographies was systematically decreased from 20 μm to 10, 5, and 2.5 µm on separate cell culture substrates. For 2D contact guidance cues, the height of the patterns was further reduced to 0 μm, whereas the width was identical to that of 2.5D contact guidance cues. (**B**) Characterization of contact guidance cues using a laser scanning profilometer shows dimensional accuracy. The 3D profile of patterned cell culture substrates H20, H10, H5, H2.5, and H0 with pattern heights of 20, 10, 5, 2.5, and 0 μm, respectively. (**C**) Cross-sectional height profile of patterned substrates, with H20 (black), H10 (blue), H5 (red), H2.5 (green), and H0 (yellow).

**Figure 2 bioengineering-11-00402-f002:**
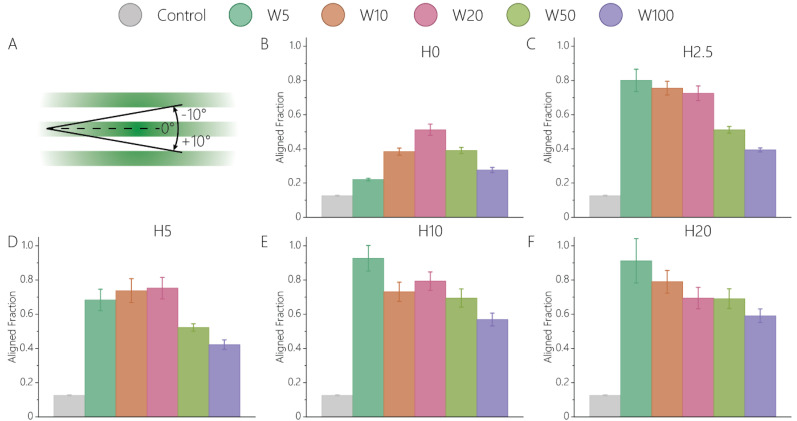
Fibroblastic keratocyte orientation is strongly influenced by the height of contact guidance cues. (**A**) The cells aligned in a 20° range around 0°, with 0° representing perfect alignment along the surface patterns, or a perfect contact guidance response. The fraction of cells aligned on (**B**) 2D cues lacking any height; 2.5D topographies with a height of (**C**) 2.5 μm, (**D**) 5 μm, (**E**) 10 μm, and (**F**) 20 μm. For each sample, the data were visualized as condition mean ± standard error of the mean. Complete (180° range) orientation distributions are given in Appendix A. Results from the statistical analysis are given in Appendix A.

**Figure 3 bioengineering-11-00402-f003:**
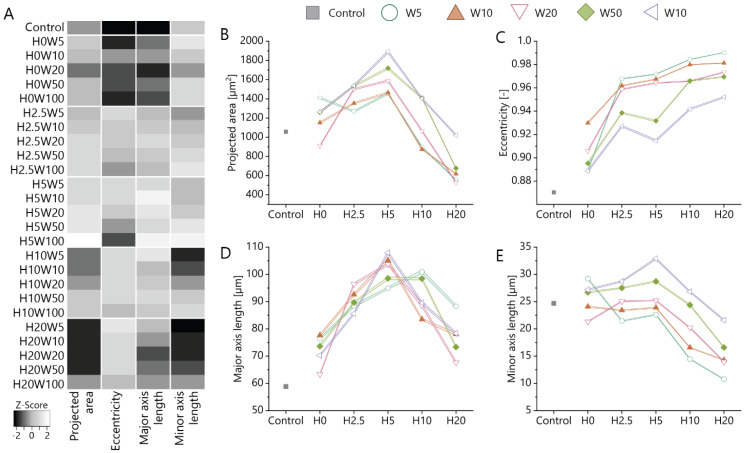
(**A**) Heatmap of morphological parameters of fibroblastic keratocytes when cultured on contact guidance cue presenting substrates. Gray scales are normalized for each readout. (**B**) The projected area, (**C**) eccentricity, (**D**) major axis length, and (**E**) minor axis length of cells on substrates with and without (control, grey) contact guidance cues. All data are presented as mean ± standard error of mean. Statistical differences between conditions are described in Appendix A.

**Figure 4 bioengineering-11-00402-f004:**
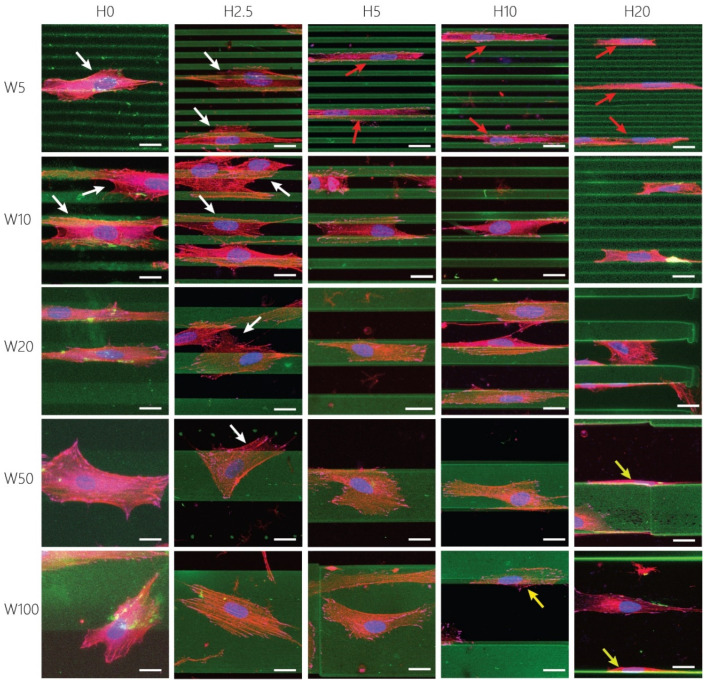
Representative maximum-intensity projections scaled to the size of single fibroblastic keratocytes on surface patterns. Visualized are the gelatin cues and coating (green), nuclei (blue), F-Actin (red), and Vinculin (magenta). White arrows indicate cells spreading across multiple cues, red arrows indicate confined cells, yellow arrows indicate cells adhering to the sides of the cues. Scale bars represent 20 μm.

**Figure 5 bioengineering-11-00402-f005:**
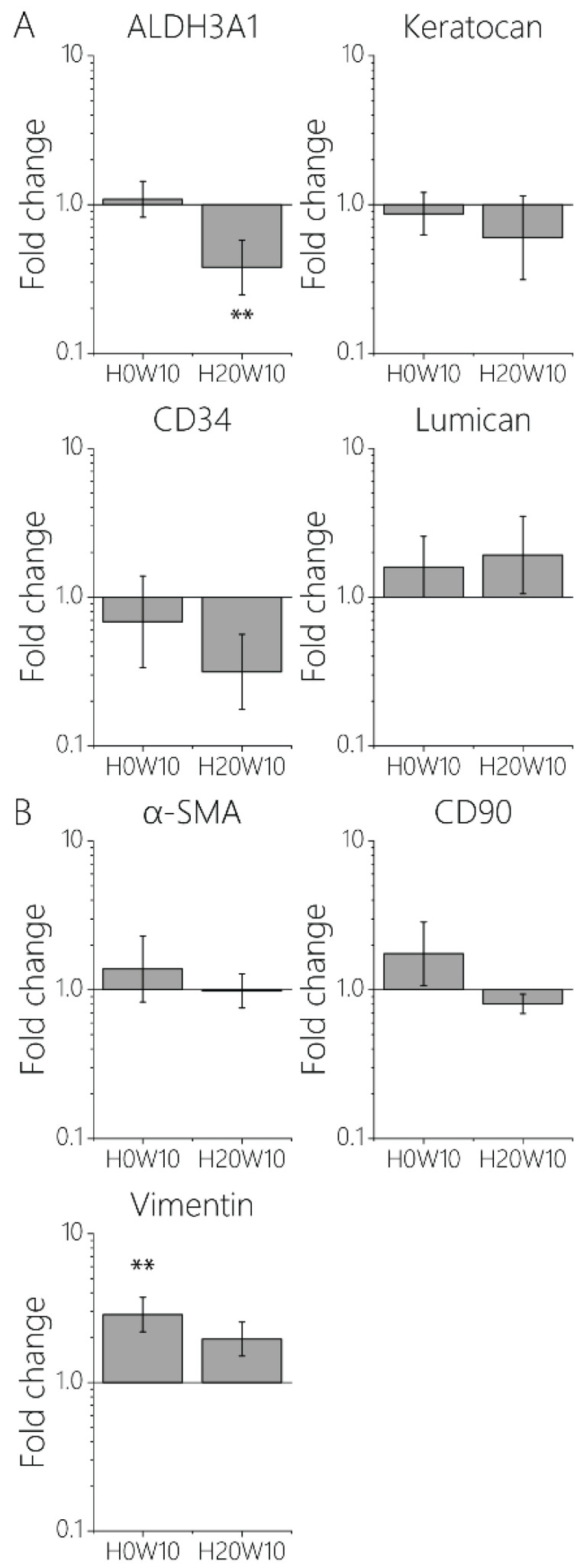
Quantified gene expression of (**A**) keratocyte markers ALDH3A1, keratocan, CD34, and lumican, and (**B**) fibroblast markers α-SMA, CD90, and vimentin with respect to fibroblastic keratocytes on flat substrates. ** *p* < 0.01.

**Figure 6 bioengineering-11-00402-f006:**
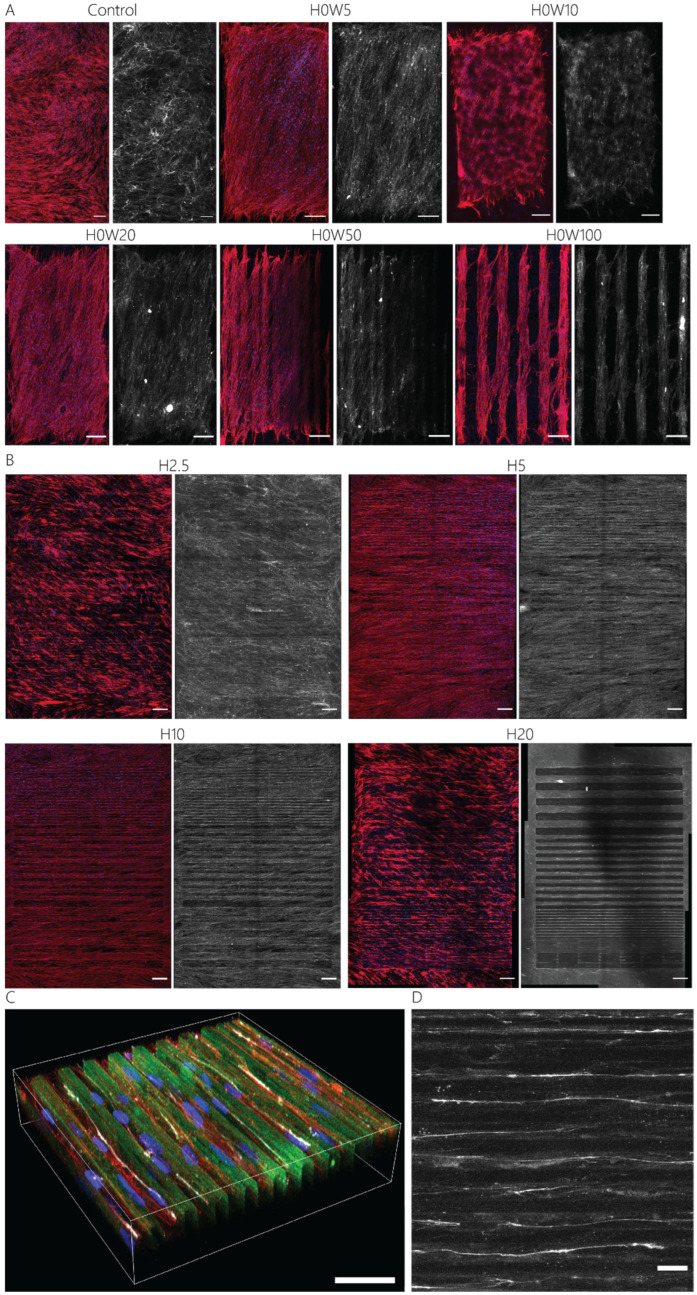
Fibroblastic keratocytes and deposited collagen after 7 days of culture on contact guidance cue presenting substrates. Maximum intensity projections show F-actin (red), nuclei (blue), and collagens (white), with (**A**) 2D; control (no pattern), 5, 10, 20, 50, and 100 μm wide lines, and (**B**) 2.5D; 2.5, 5, 10, and 20 μm tall and 5, 10, 20, 50, and 100 μm wide contact guidance cues. Scale bars represent 200 μm. (**C**) Representative 3D visualization of the contact guidance cues (H20W10) with cells and collagen mainly in between the cues. Visualized are F-actin (red), nuclei (blue), gelatin (green), and collagens (white). Scale bar represents 40 μm. (**D**) Maximum-intensity projection of a zoomed-in region of the cues, showing the location and alignment of newly deposited collagen (white). Scale bar represents 20 μm.

**Figure 7 bioengineering-11-00402-f007:**
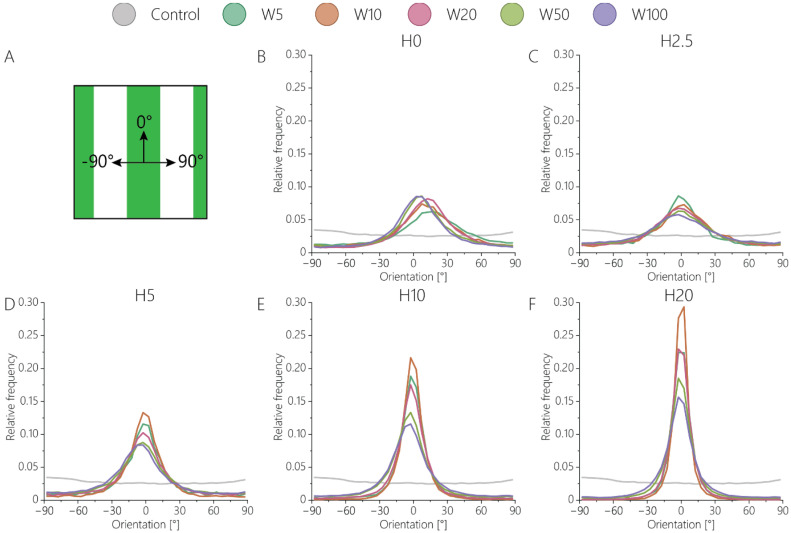
The orientation distribution of collagen with respect to the contact guidance cues. (**A**) 0° represents collagen aligning according to the cue direction, whereas −90° represents a counter-clockwise rotation, and 90° represents a clockwise rotation. Orientation distributions of (**B**) 2D contact guidance cues, and 2.5D topographies with heights of (**C**) 2.5 μm, (**D**) 5 μm, (**E**) 10 μm, (**F**) 20 μm.

**Table 1 bioengineering-11-00402-t001:** Dimensions of the various contact guidance cues and accompanying nomenclature.

Width (μm)	Height (μm)				
	2D	2.5D			
	0	2.5	5	10	20
0	Control				
5	H0W5	H2.5W5	H5W5	H10W5	H20W5
10	H0W10	H2.5W10	H5W10	H10W10	H20W10
20	H0W20	H2.5W20	H5W20	H10W20	H20W20
50	H0W50	H2.5W50	H5W50	H10W50	H20W50
100	H0W100	H2.5W100	H5W100	H10W100	H20W100

## Data Availability

All experimental data within the article are available from the corresponding author upon reasonable request.

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
