# Peer review of "Dimensionality Matters: Exploiting UV-Photopatterned 2D and Two-Photon-Printed 2.5D Contact Guidance Cues to Control Corneal Fibroblast Behavior and Collagen Deposition"

_bioengineering, 2024, doi:10.3390/bioengineering11040402_

Round 1

Reviewer 1 Report

Comments and Suggestions for Authors

This paper discusses the importance of restoring native cells and extracellular matrix organization in tissues after disease or injury. They have focused on the corneal tissue as a model. The authors report the alignment of keratocytes in response to the physical microenvironment and use anisotropic contact guidance cues to engineer cornea-like tissues. Herein, two main challenges have been identified, viz., optimizing cellular contact guidance and understanding its role in long-term extracellular matrix organization. To address these challenges, the authors combined UV-based protein patterning and two-photon polymerization to create an array of anisotropic contact guidance cues with varying heights and widths. They found that both the width and height of the cues influenced short-term cell orientation and morphology. Further, the height also affected long-term collagen deposition and alignment. The authors have concluded in this study that the dimensionality of the microenvironment can guide cellular organization and tissue formation in a tailored way.

This is a very well-crafted manuscript. However, I have the following advice, which should be incorporated before the manuscript can be accepted.

1.      The resolution of the graphical abstract should be improved. Currently, many writings are not legible.

2.      Avoid using personal pronouns such as "I" or "we" in the paper.

3.      Incorporate statistical significance analysis data directly into the Figure instead of including it as supplementary information.

4.      The quality of graphs in Figure 3 is very poor. The resolution of the Figure should be improved.

5.      A study on the analysis of the functional properties of the engineered tissues should be performed to evaluate their suitability for corneal tissue engineering applications.

Comments on the Quality of English Language

Minor editing of English language required

Reviewer 2 Report

Comments and Suggestions for Authors

Using protein patterning and 2photon based cross linking strategies the authors develop dimensionally distinct scaffolds that can modulate the properties of cellular adhesion and proliferation. This is important and can be used for therapeutic applications like the corneal tissue development. Using different methods the authors elucidate how the layering and patterning of the scaffolds can affect cellular properties like adhesion, contacts, proliferation, etc. The manuscript is nicely written and in simple language to understand. 

However, few changes need to made before the decision could be reached 

1. the title of the article can be changed to include the methods and applications 

2. in the main figures of the paper, figures 4-6 the data is not at all quantified. this is very important factor and only images cannot provide any quantitative information. I would suggest the authors to quantify the data and show the statistical significance for all the data. 

3. the quantification and statistical significance should be included in the revised manuscript and proper discription of it in the discussion and conclusions section should be included. 

Comments on the Quality of English Language

English language is overall ok. please check the manuscript through the software like Grammarly to make it professionally eligible.

Round 2

Reviewer 2 Report

Comments and Suggestions for Authors

The authors have adequately answered my queries and the revised manuscript can now be recommended for publication